# Attention to the strengths of physical interactions: Transformer and graph-based event classification for particle physics experiments

Luc Builtjes[1], Sascha Caron[1,2*], Polina Moskvitina[1,2†], Clara Nellist[2,3‡], Roberto Ruiz de Austri[4°], Rob Verheyen[5§] and Zhongyi Zhang[1,2,6¶]

**1** High Energy Physics, Radboud University Nijmegen, Heyendaalseweg 135, 6525 AJ Nijmegen, the Netherlands
**2** Nikhef, Science Park 105, 1098 XG Amsterdam, the Netherlands
**3** Institute of Physics, University of Amsterdam, 1090 GL Amsterdam, The Netherlands
**4** Instituto de Física Corpuscular, IFIC-UV/CSIC, Valencia, Spain
**5** Department of Physics and Astronomy, University College London, London, WC1E 6BT, UK
**6** The Bethe Center for Theoretical Physics, Bonn University, 53115 Bonn, Germany

⋆ scaron@nikhef.nl , † p.moskvitina@nikhef.nl , ‡ c.nellist@nikhef.n ,
∘ rruiz@ific.uv.es , § r.verheyen@ucl.ac.uk , ¶ zhongyi@th.physik.uni-bonn.de

## Abstract

A major task in particle physics is the measurement of rare signal processes. Even modest improvements in background rejection, at a fixed signal efficiency, can significantly enhance the measurement sensitivity. Building on prior research by others that incorporated physical symmetries into neural networks, this work extends those ideas to include additional physics-motivated features. Specifically, we introduce energy-dependent particle interaction strengths, derived from leading-order SM predictions, into modern deep learning architectures, including Transformer Architectures (Particle Transformer), and Graph Neural Networks (Particle Net). These interaction strengths, represented as the SM interaction matrix, are incorporated into the attention matrix (transformers) and edges (graphs). Our results in event classification show that the integration of all physics-motivated features improves background rejection by $10\% - 40\%$ over baseline models, with an additional gain of up to 9% due to the SM interaction matrix. This study also provides one of the broadest comparisons of event classifiers to date, demonstrating how various architectures perform across this task. A simplified statistical analysis demonstrates that these enhanced architectures yield significant improvements in signal significance compared to a graph network baseline.

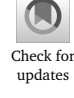

# 1   Introduction

With the unprecedented amount of data provided by the upcoming runs of the Large Hadron Collider (LHC), one can start to measure rare processes with very small cross-sections. Notable examples include the recent observations of 4-top quarks originating from a single proton-proton collision event [1, 2]. At the heart of this endeavor is the difficult task of detecting and measuring rare signal processes amidst the overwhelming background noise generated by the multitude of Standard Model (SM) processes. The accurate classification of these events is crucial, as even a small reduction in background noise—on the order of a few tens of per-cent—while maintaining the same signal detection efficiency can lead to a profound increase in sensitivity.

Boosted decision trees (BDTs) have long been used as the standard for event-level classification in LHC experiments, often employing high-level features derived from domain expertise to enhance performance [3–5]. However, there has been growing interest in applying deep learning architectures to this task. While methods such as Convolutional Neural Networks (CNNs), Graph Neural Networks (GNNs), and attention-based models have achieved significant success in the classification of jet data [7, 10–14], their application to event-level classification has not been explored to the same extent. Recent studies have begun to demonstrate the effectiveness of neural networks for event-level classification tasks [6, 8, 9].

On the other hand, recent advances have seen the incorporation of physical principles directly into machine learning (ML) processes, thereby improving the performance and interpretability of these models. For instance, several studies have shown that embedding Lorentz symmetry into ML models for tasks such as jet tagging can significantly enhance their effectiveness in particle physics applications [15–19]. However, exploring the embedding of Lorentz symmetry and other physical symmetries directly at the event level remains an open challenge and is addressed in this work.

In this work, we build on previous efforts to incorporate physical symmetries into machine learning architectures for event classification in high-energy physics (HEP) by integrating energy-dependent particle-particle interaction strengths, as predicted by the leading-order interactions of the Standard Model. This approach expands on the physical information previously proposed for jet tagging by including pairwise kinematic features that capture particle-particle interaction strengths (detailed in Chapter 4). These features are embedded in advanced classification methods, including Boosted Decision Trees, Transformer architectures (such as Particle Transformer), and Graph Neural Networks (such as ParticleNet). Furthermore, we explore the inclusion of Lorentz symmetry in event-level classification, which has not been done before. To facilitate model testing, we present a publicly available dataset with 4-top and top-top-Higgs events. Wide hyperparameter scans are conducted across all models, and performance metrics are computed to evaluate their predictive capabilities.

In Section 2, the data generation process and data format are described. Section 3 provides a brief overview of the machine learning models used in the comparison and their optimization. In Section 4, the question of how to inform the ML models about physics is explored. Results are presented and discussed in Section 5, followed by conclusions in Section 6.

## 2 Data description

In this section, the data generation and the data format used for this work are described.

### 2.1 Data generation

Proton-proton collisions at a center-of-mass energy of 13 TeV were simulated. The relevant production processes for this work consist of the backgrounds $t\bar{t} + \mathrm{X}$, where $X = Z, W^+$, $W^+W^-$ and signal processes including the 4-top production process and $t\bar{t}h$ production. The hard scattering process generation was performed at leading order, where up to two additional jets are added to the final state in the case of the background processes, and up to one for the signal. The cross-sections for all the processes and the corresponding total number of events generated are depicted in Table 1.

The hard scattering events were generated using `MG5_aMC@NLO` version 2.7 [20] with the `NNPDF31_lo` parton distribution function set [21], utilizing the 5-flavor scheme. Parton showering was performed with `Pythia` version 8.239 [22], while the MLM merging scheme [23] was used to merge high-multiplicity hard scattering events with the parton shower. A fast

Table 1: Signal and background processes with the corresponding LO (Leading Order) cross-section $\sigma$ in pb (second column), the total number of generated events $N_{\text{tot}}$ (third column), and the efficiency $\epsilon$ of the cuts applied (fourth column).

| Physics process | $\sigma$ (pb) | $N_{\text{tot}}$ | $\epsilon$ |
|---|---|---|---|
| $pp \to t\bar{t}t\bar{t}$ (+1 j) | 0.01 | 32463742 | 0.007 |
| $pp \to t\bar{t}h$ (+2 j) | 0.022 | 29783343 | 0.001 |
| $pp \to t\bar{t}W^{\pm}$ (+2 j) | 0.045 | 8954246 | 0.005 |
| $pp \to t\bar{t}W^{+}W^{-}$ (+2 j) | 0.0096 | 20160377 | 0.003 |
| $pp \to t\bar{t}Z$ (+2 j) | 0.034 | 10605846 | 0.011 |

detector simulation was performed using `Delphes` version 3.4.2 [24] with the ATLAS detector map. Finally, an $H_T = \sum_{jets} E_T > 400$ GeV restriction was imposed at the parton level during event generation. The purpose of this restriction is to enhance generation efficiency, as the signal region imposes $H_T > 500$ GeV on the reconstructed objects (see Section 2.2 for details.).

## 2.2 Event and object selection

Collision events consist of objects such as jets, b-jets, leptons, and photons, each with their corresponding kinematic variables (see Section 2.3). Following the strategy in Ref. [25], an event was saved if at least one of the following conditions was met:

- At least one jet or $b$-jet with transverse momentum $p_{\text{T}} > 60$ GeV and pseudorapidity $|\eta| < 2.8$. The $b$-jets were defined using the ATLAS Delphes default card, which is based on truth-level (gen-level) matching to partons, as discussed in [26].

- At least one electron with $p_{\text{T}} > 25$ GeV and $|\eta| < 2.47$, except for $1.37 < |\eta| < 1.52$.

- At least one muon with $p_{\text{T}} > 25$ GeV and $|\eta| < 2.7$.

- At least one photon with $p_{\text{T}} > 25$ GeV and $|\eta| < 2.37$.

The subsequent object selection followed the criteria outlined in Ref. [27], whereby individual objects were kept only if they passed the following requirements:

- Electron candidates with $p_T > 28$ GeV and $|\eta| < 2.47$ were selected. Electrons in the region $1.37 < |\eta| < 1.52$, known as the LAr crack region, were rejected to reduce contributions from non-prompt and fake electrons due to the detector design in the liquid Argon calorimeter.

- Muon candidates were required to pass the Medium quality working point, with $p_T > 28$ GeV, and $|\eta| < 2.5$.

- Jet candidates with $p_T > 25$ GeV and $|\eta| < 2.5$ were selected.

To minimize background as much as possible with respect to the signal events, a signal region [27] was defined. This region required at least six jets, at least two of which were b-tagged, $H_T > 500$ GeV, and two same-sign leptons or at least three leptons for each event. The resulting efficiencies are shown in Table 1.

## 2.3 Data format

The generated Monte Carlo data were saved as ROOT files and processed into CSV files using the event selection presented in Section 2.2. Each line in the CSV files has a variable length and contains three event-specifiers, followed by the kinematic features for each object in the event. The specific line format follows the event structure used in the Dark Machines challenges [25, 28], and is given by:

$$\text{event ID; process ID; weight; } \not{E}_T; \ \phi_{\not{E}_T}; \ \text{obj}_1, E_1, p_{T_1}, \eta_1, \phi_1; \ \text{obj}_2, E_2, p_{T_2}, \eta_2, \phi_2; \cdots,$$

where each object is represented by a string starting with an identifier obj_n,[1] followed by its kinematic properties in the form of a four-vector containing the total energy $E$, transverse momentum $p_T$ in units of MeV, pseudorapidity $\eta$, and azimuthal angle $\phi$. The other relevant quantities are $\not{E}_T$ and $\phi_{\not{E}_T}$, which represent the magnitude of the missing transverse energy $E_T^{\text{miss}}$ and its azimuthal angle $\phi_{E_T^{\text{miss}}}$. The other three components represent the identity of an event, the corresponding physical process, and the event weight, which is given by the cross-section of the process divided by the total number of generated events.

The event generation was performed in two steps. Initially, a smaller sample of about **50,000** events was generated, serving as the baseline for this analysis and providing a dataset comparable in size to what is typically available in ATLAS studies. Subsequently, additional events were generated to investigate the scaling of model performance with increasing training data, allowing us to assess whether performance saturates as the dataset size grows.

Since the length of the events is variable, the data were zero-padded to match the largest number of objects found across all events in the dataset. To create a balanced dataset between the 4-top signal and background processes, **150,000** signal events corresponding to the 4-top process were selected. Approximately **30,000** events were generated for the $t\bar{t}h$ process due to its low acceptance rate, and around **40,000** events were generated for each of the other background processes, resulting in a total of approximately **150,000** background events. Therefore, the total number of events used in the dataset was **302,072**. The dataset was split into **80%** for training, **10%** for validation, and **10%** for testing. All background processes contributed equally to the background portion of the dataset. The data are publicly available in CSV format at Ref. [29].

# 3 Machine learning models

In this section, we provide a brief summary of the models and methods used in this work.

## 3.1 Boosted decision tree

While it is common in HEP analyses to design high-level features to enhance the performance of traditional classifiers like BDTs [30–32], the approach presented in this work uses the raw event-level data directly, specifically the four-momentum components (4-vectors) of particles. Lower-level detector information, such as hits or calorimeter clusters, was not included. To evaluate the performance of BDTs in this event-level classification task, we employ Light Gradient Boosting Machine (LightGBM[2]).

BDTs combine a series of weak classifiers (decision trees) into a stronger classifier through gradient boosting. The boosting strategy is defined with respect to a series of previous decision

---

[1]j: jet, b: b-jet, e-: electron, e+: positron, $\mu$-: muon, $\mu$+: antimuon, g: photon.

[2]http://github.com/microsoft/LightGBM with `binary cross-entropy` as the loss function, `AUC` as the early stopping metric, 5000 estimators, 500 leaves, a learning rate of 0.01, the `gbdt` boost type, and a maximum depth of 15. For details, refer to Appendix B.1.

trees, $f_1, f_2, \cdots, f_{t-1}$ which remain fixed while the $t$-th tree $f_t$ is calculated. This process is made highly efficient in LightGBM by converting the input data into histograms and using gradient-based sampling to focus on the data that are not well modelled. This procedure reduces memory usage and is optimized for both CPU and GPU performance. LightGBM uses first- and second-order derivatives to minimize the loss for the next iteration in the gradient boosting process:

$$
\begin{aligned}
\text{Loss}^{(t)} &= \sum_{i=1}^{n} l(y_i, (\hat{y}_i^{(t-1)} + f_t(x_i))) + \sum_{i=1}^{t} \omega(f_i) \\
&\approx \sum_{i=1}^{n} [g_i f_t(x_i) + \frac{1}{2} h_i f_t^2(x_i)] + \omega(f_t) + \text{constant}, \\
g_i &= \partial_{\hat{y}_i^{(t-1)}} l(y_i, \hat{y}_i^{(t-1)}), \\
h_i &= \partial^2_{\hat{y}_i^{(t-1)}} l(y_i, \hat{y}_i^{(t-1)}).
\end{aligned}
\tag{1}
$$

Here, $l$ represents the reconstruction loss functions (e.g., Mean Square Error, Binary Cross Entropy), $f_t$ is the $t$-th tree, and $\hat{y}^{(t-1)}$ is the class label predicted by trees $f_1, f_2, \cdots, f_{t-1}$. The term $\omega(f_i)$ represents tree complexity terms that involve properties such as depth and number of leaves. LightGBM uses a depth-first algorithm to add branches to the tree $f_t$ with a limitation on the maximum depth while minimizing Equation (1).

A common property of collision data in particle physics is a wide variability in the number and types of objects in an event. A structured format of such data typically leads to a high degree of sparsity, which we found to significantly degrade the performance of BDTs. Therefore, the data were pre-processed by limiting the maximum number of (jets, b-jets, $e^-$, $e^+$, $\mu^-$, $\mu^+$) to the $(4, 4, 1, 1, 1, 1)$ hardest objects, respectively.

One notable advantage of BDTs is the significantly faster training time compared to other architectures described in this paper. This makes hyperparameter fine-tuning and data format adjustments much more efficient. Another key feature of BDTs is their ability to indicate feature importance, which provides insight into which input variables contribute most to the model's performance. In Section 4.1, the inclusion of high-level features beyond the raw four-vectors is discussed, where this feature is especially useful.

## 3.2 Fully connected network

Fully connected neural networks (FCNs) [33] are one of the most basic forms of deep neural networks. They consist of multiple layers of neurons, where each neuron in a layer is connected to every neuron in the subsequent layer. These connections represent a linear transformation with trainable parameters, followed by a non-linear activation function. After the final layer, which consists of a single node, a sigmoid activation function is applied to produce a classification score.

All hidden layers in the FCN use ReLU activation functions. Additionally, `Dropout` [34] with a probability 0.5 is applied after the first three hidden layers to prevent overfitting. The network is trained using the Adam optimizer [35] with default parameters, which include $\beta_1 = 0.9$, $\beta_2 = 0.999$, and $\epsilon = 1e^{-7}$. Here, $\beta_1$ and $\beta_2$ control the exponential decay rates for the first and second moment estimates of the gradients, while $\epsilon$ is a small constant added to prevent division by zero. The learning rate, however, was treated as a hyperparameter and optimized separately.

The hyperparameters optimized for the FCN, including the batch size, learning rate, number of layers, and number of neurons per layer, are provided in Appendix B.2. Early stopping is applied by monitoring the performance on the validation set.

Table 2: Particle input variables for the 1D CNN, ParticleNet and Particle Transformer. FCNs and BDTs use the same variables, but limit the information to the four-momentum components. The variable $\gamma_{\text{tag}}$ refers to the identification tag assigned to photons in the event. For leptons, only the one lepton with the highest $p_T$ per lepton type (including charge) is considered, while for jets and $b-$jets, only the four jets with highest $p_T$ are included. As a result, only 12 objects are used for FCNs and BDTs. For BDTs, MLPs, and CNNs, the information is ordered by the $p_T$.

| Variables per particle |
| --- |
| E, $p_T$, $\eta$, $\phi$, jet$_{\text{tag}}$, b-jet$_{\text{tag}}$, $e^-_{\text{tag}}$, $e^+_{\text{tag}}$, $\mu^-_{\text{tag}}$, $\mu^+_{\text{tag}}$, $\gamma_{\text{tag}}$ |

Similar to the case for BDTs, the performance of the FCN was found to generally deteriorate when applied to large, sparse input data. Therefore, the data were pre-processed following the prescription given in Section 3.1.

## 3.3 Convolutional network

Convolutional Neural Networks (CNNs) [36] are primarily applied to analyze data where adjacent elements have a causal relationship, such as in image data. CNNs use convolutional operations through trainable filter matrices that slide over the data, producing outputs that are translationally equivariant. The convolutional layers are typically followed by pooling operations to reduce the dimensionality of the data within the network layers. In this work, we utilize a one-dimensional variant of CNN architecture (1D CNN), inspired by the DeepAK8 algorithm, which was originally used for jet tagging [37].

We incorporate 11 particle features, as given in Table 2. Each feature is represented by an array of size $N_{\text{max}} = 18$, which corresponds to the maximum number of objects in an event. The event-wide features $E_T^{\text{miss}}$ and $\phi_{E_T^{\text{miss}}}$ are added to the $p_T$ and $\phi$ feature vectors, respectively. Following [37], the network consists of a set of 1D convolutional blocks that process each feature vector independently. The outputs from these blocks are concatenated and passed to a fully connected network (FCN) with ReLU activations. Each convolutional block is composed of two sub-blocks. These sub-blocks include a set of convolutional layers, each with a ReLU activation function, followed by a `MaxPooling` layer and a `Dropout` layer with a dropout probability 0.2 to prevent overfitting. The final layer of the model uses a softmax activation function to output class probabilities for binary classification.

The model was trained using the Adam optimizer with default parameters, and categorical cross-entropy was used as the loss function. Early stopping was applied, monitoring the validation AUC (Area Under the Curve) to avoid overfitting. The learning rate and all other hyperparameters, including the number of convolutional layers, the number of filters, kernel sizes, and fully connected network parameters, are provided in Appendix B.3.

## 3.4 ParticleNet

ParticleNet (PN) [38] is a graph-based architecture based on Dynamic Graph Convolutional Neural Networks [39]. Events are treated as particle clouds, inspired by point clouds [39] used in Computer Vision challenges. Each final-state particle, represented by the variables shown in Table 2, is encoded as an individual node in the graph. The node features consist of the particle's kinematic variables ($E$, $p_T$, $\eta$, $\phi$) and a one-hot encoded vector representing the particle type. This ensures that both the kinematic information and particle identity are included in the representation.

Edges in the graph are constructed dynamically by connecting each particle to its $k$-nearest neighbours (kNN), with distances in angular space defined as $\Delta R_{ij} = \sqrt{(\Delta \eta)_{ij}^2 + (\Delta \phi)_{ij}^2}$. The graph representing the event thus has $N$ nodes (corresponding to the number of final state particles) and $kN$ edges.

The node features are updated through a message-passing operation defined as:

$$x_i' = \frac{1}{k} \sum_{j \in \mathcal{N}(i)} \text{FCN}(x_i, x_j - x_i), \tag{2}$$

where $\mathcal{N}(i)$ is the set of $k$-nearest neighbors for node $i$, and FCN is a fully connected network that rocesses both the node features $x_i$ of the central particle and the relative differences $x_j - x_i$, enabling the model to learn relational information between particles. The resulting features from all neighbors are aggregated using mean aggregation, ensuring permutation invariance across the neighbors.

The updated node features $x_i'$ are passed through multiple layers of this operation. Features from all layers are concatenated, averaged over the nodes, and combined with additional event-level features such as $E_T^{\text{miss}}$ and $\phi_{E_T^{\text{miss}}}$. These final features are processed by another FCN to produce the final classification.

To further improve performance, an attention-weighted aggregation mechanism was explored, replacing the simple mean aggregation with a weighted sum. In this approach, attention weights are learned through trainable transformations of the features, allowing the model to assign different importance levels to messages from neighboring particles. However, no significant performance improvements were observed compared to the simpler mean aggregation for the event-level classification task.

A wide hyperparameter scan was performed on the PN architecture, and no significant performance differences were found as long as sufficient model capacity was available. Therefore, the hyperparameter settings recommended in Ref. [38] were chosen, and the training procedure from Ref. [19] was followed. Our implementation is based on Ref. [40]. The hyperparameters are detailed in Appendix B.4.

## 3.5 Particle Transformer

Particle Transformer (ParT) [10] is a transformer-based architecture originally developed for jet tagging. Inspired by the success of similar architectures in fields such as natural language processing [41], it treats individual particles as embedding rather than words. At its core lies the repeated application of the self-attention mechanism:

$$\text{Attention}(Q, K, V) = \text{SoftMax}\left(QK^T / \sqrt{d}\right) V, \tag{3}$$

where $Q$, $K$ and $V$ are trainable $d$-dimensional linear projections derived from the same particle embeddings, based on the variables listed in Table 2. The query ($Q$) encodes the current state of each particle, the key ($K$) represents reference points for comparing particles, and the value ($V$) contains particle features that are updated based on the attention weights. This process allows the model to refine each particle's representation by leveraging relationships between all particles in the event.

This approach allows the model to compute attention scores between particles by taking the dot product of the query and key matrices, normalized by the embedding dimension $d$, and then applying a softmax to compute attention weights. These attention weights are then used to update the value matrix $V$, effectively refining each particle's representation based on the relationships between particles in the event.

The application of the attention mechanism serves to correlate each particle with all others. Furthermore, it is applicable to vary numbers of particles and is explicitly permutation-*invariant*. Although the self-attention mechanism itself is *equivariant* to permutations, its implementation within the Particle Transformer ensures permutation-*invariance* by treating the input as an unordered set and processing particle interactions based on their features, ensuring the output is unaffected by input order.

Classification is obtained by appending a classification token to the list of particle embeddings before the final layers of the transformer. This token is a trainable set of weights that is identical for every event. The attention mechanism then correlates the classification token with the event, after which it is processed by an FCN, which also receives $E_T^{\text{miss}}$ and $\phi_{E_T^{\text{miss}}}$, to produce a classification label. Our implementation is based on Ref. [42].

As with ParticleNet, a hyperparameter scan over the ParT architecture did not result in significant differences in performance. Therefore, the hyperparameter settings and training procedure recommended in Ref. [10] were applied for this study. The hyperparameters are detailed in Appendix B.5.

### 3.6 Particle Transformer as Set Transformer

The Set Transformer architecture [43] was also incorporated into the ParT model. In a Set Transformer, the matrix $Q$ in Eq. (3) is no longer a projection of the input states but is instead composed a set of so-called inducing points. These inducing points, a set of learnable parameters, are jointly optimized with the rest of the transformer's parameters. Designed to provide a reduced, latent representation of the input data, the inducing points enable the model to efficiently summarize the information contained in $V$ for any possible input state.

The model is trained to use these inducing points to represent the input data, replacing the direct projection of input states. This approach leads to a more compact and computationally efficient attention mechanism. The scaled dot-product attention mechanism is thus modified as follows:

$$\text{Attention}(Q', K, V) = \text{SoftMax}\left(Q'K^T / \sqrt{d}\right) V. \tag{4}$$

Here, $Q'$ represents the inducing points (a set of learned vectors). These points act as a summary or a learned representation of your input set, allowing the attention mechanism to focus more efficiently on specific aspects of the data. $K$ and $V$ are derived from the input set.

Unlike standard self-attention, which is permutation-*equivariant*, this modification leads to a self-attention mechanism that is permutation *invariant* [44], meaning that permuted inputs produce exactly the same output. Since collision data presents as an unordered set of particles, the performance of the model may benefit from the former, as it imposes a stricter constraint.

Several configurations for the number of inducing points were tested, with sets of {18, 20, 30, 40, 50, 100, 200} points being explored. While the performance of the transformer model without pairwise features increased slightly with increasing number of inducing points, we did not observe any improvement with increasing number of inducing points in the model with pairwise features. Once again, the best Set Tranformer model proved to be the one that incorporated all pairwise kinematic interactions and SM running coupling constants represented by the third SM interaction matrix, as explained in the following chapter (labelled 'SetT$_{\text{int. SM}}$').

### 3.7 Particle Transformer with focal loss

For the particle transformer, experiments were conducted using the focal loss [45] as a replacement for the standard binary cross-entropy (BCE) loss. Focal loss is particularly effective in addressing class imbalances by emphasizing hard-to-classify examples while reducing the influence of well-classified ones.

The BCE loss, commonly used for classification tasks, is defined as:

$$\text{BCE} = -\frac{1}{N} \sum_{i=1}^{N} \left[ y_i \log(p_i) + (1 - y_i) \log(1 - p_i) \right], \tag{5}$$

where $N$ is the number of samples, $y_i \in \{0, 1\}$ is the true label, and $p_i$ is the predicted probability for the positive class. While BCE loss is effective in general, it can struggle with class imbalance, as the majority class often dominates the loss and minority classes may be underrepresented.

Focal loss addresses this by introducing a scaling factor $(1 - p_i)^\gamma$, which reduces the contribution of well-classified examples and focuses on harder ones:

$$\text{Focal Loss} = -\frac{1}{N} \sum_{i=1}^{N} \alpha_t (1 - p_i)^\gamma \log(p_i), \tag{6}$$

where:

- $\alpha_t$ is a class balancing factor, defined as:

$$\alpha_t = \begin{cases} \alpha, & \text{if } y_i = 1, \\ 1 - \alpha, & \text{if } y_i = 0, \end{cases}$$

  with $\alpha$ chosen based on class distribution.

- $\gamma$ is the focusing parameter, which controls the rate at which easy examples are downweighted. Larger $\gamma$ values place greater emphasis on hard-to-classify examples.

- $p_i$ is the predicted probability for the positive class.

The scaling term $(1 - p_i)^\gamma$ effectively suppresses the loss contribution from well-classified examples (e.g., $p_i$ close to 1 for positive samples or close to 0 for negative samples) while amplifying the contribution of misclassified ones. Typical values for $\gamma$ range from 0 to 5, and larger values place greater emphasis on harder examples. Similarly, $\alpha$ adjusts the class weights to ensure that minority classes are adequately represented.

A comprehensive hyperparameter scan was performed over the focal loss parameters using the extended ParT model, which includes pairwise kinematic features and SM running coupling constants represented by the third SM interaction matrix (discussed in the following chapter). The scan explored values of $\alpha \in \{0.25, 0.5, 0.75, 1\}$ and $\gamma \in \{0, 1, 2, 3, 4, 5, 6\}$. The optimal parameters were found to be $\alpha = 0.75$ and $\gamma = 3$, resulting in the model labelled as 'ParT$_{\text{int. SM (FL)}}$'.

However, even with these optimal settings, the overall performance of the model was not better than that of models trained with the standard cross-entropy loss. Thus, results presented below refer to models trained with cross-entropy loss, unless otherwise specified. Nonetheless, the focal loss showed improved performance in separating specific background processes, suggesting that its utility may be context-dependent and better suited for certain background distributions.

## 4 Informing the ML models about physics

### 4.1 Pairwise kinematic features based on four-vectors

Previous studies have demonstrated that the inclusion of additional information beyond raw four-vector data, such as correlations between four-vectors (referred to here as pairwise features), can improve the performance of deep learning classifiers in jet physics [10, 19]. These

pairwise four-vector correlations typically include invariant masses or distances between two particles, which are commonly used in jet physics.

In our analysis, a distinction is made between **covariant representations** and **physics information**. Covariant representations consist of features that respect the fundamental symmetries of the physical system, such as Lorentz invariance, ensuring that quantities like invariant mass remain unchanged under Lorentz transformations. In contrast, physics information encompasses additional insights into particle interactions, such as coupling constants and interaction strengths derived from the SM, which will be introduced in Section 4.2.

For the BDT, experiments were performed with the inclusion of various high-level features, which were treated on the same footing as low-level ones. Similarly, following Refs. [10, 19], pairwise features were included in PN and ParT through a trainable embedding $U_{ij}$ for particles $i$ and $j$. These embeddings were incorporated into PN by modifying Equation (2) as:

$$x'_i = \frac{1}{k} \sum_{j \in \mathcal{N}(i)} \text{FCN}(x_i, x_j - x_i + U_{ij}), \tag{7}$$

and into ParT by replacing Equation (3) with:

$$\text{Attention}(Q, K, V) = \text{SoftMax}\left(QK^T / \sqrt{d} + U\right) V. \tag{8}$$

Here, $Q$ (query) and $K$ (key) matrices are derived from the particle embeddings, and their dot product $QK^T$ represents the interaction or similarity between particles. This determines how much attention each particle pays to others. The pairwise feature matrix $U$ captures additional kinematic information about particle pairs, such as invariant masses and angular distances. Initially, $U$ is a three-dimentional matrix of shape $N \times N \times F$, where $N$ is the number of particles and F is the number of pairwise features. To align the dimentionality of $U$ with $QK^T$, a 1D convolution is applied along the $F$-axis, reducing $U$ to a two-dimensional matrix of shape $N \times N$. This operation ensures compatibility between $U$ and $QK^T$, allowing them to be summed element-wise. The resulting attention scores are then used to weight the values $V$, producing updated particle representations that account for both individual features and pairwise relationships.

In all three above cases, the performance of a wide variety of kinematic pairwise features was evaluated, including $m_{ij}$, $\Delta R_{ij}$, the jet-based features used in Ref. [10] and three-body invariant masses. Two-body invariant masses $m_{ij}$ were calculated, and this was extended to three-body systems by adding the hardest and second-hardest particles in the event, denoted as $m_{ij,1}$ and $m_{ij,2}$. However, these features (including the hardest, second hardest, and more) did not significantly improve the performance in ParticleNet, Transformer, or BDT models. Using the feature importance indicator of the BDT, and empirically for ParticleNet and Particle Transformer, it was found that for all architectures the performance was saturated by the inclusion of only $m_{ij}$ and $\Delta R_{ij}$. Furthermore, the BDT results also indicated that pairwise invariant masses provided the biggest gain in performance. This result is in line with the findings of [19]. In the next section, experiments were conducted by adding further information through dynamics, while maintaining the kinematic information described above.

## 4.2 Pairwise kinematic features and the Standard Model interaction matrix

The Standard Model (SM) of particle physics provides the most comprehensive framework for understanding the electromagnetic, weak, and strong nuclear interactions between elementary particles. In this work, the dynamics of particle interactions described by the SM are explored and incorporated through the inclusion of a separate interaction matrix in the embedding $U_{ij}$ for the PN and ParT models. The interaction matrix consists of entries that indicate the significance of pairwise particle interactions. To systematically investigate the

effect of adding dynamic information to the models, three types of interaction matrices with increasing amounts of physical information are explored.

In the first matrix (abbreviated as SMids and referred to as SM matrix[1] in Table 3), an entry of '1' indicates an interaction possible at the leading order in the SM, while a '0' signifies interactions that only appear at higher orders. The following pairwise interactions are assigned a '1' in the matrix: jet–jet, jet–b-jet, jet–$\gamma$, b-jet–b-jet, b-jet–$\gamma$, $e^- - e^+$, $e^- - \gamma$, $e^+ - \gamma$, $\mu^- -\mu^+$, $\mu^- -\gamma$, $\mu^+ -\gamma$. The omission of other particle interactions (marked with a '0') does not imply that they are physically impossible, but rather represents a practical limitation for the model. While the simplified representation does not take into account the full complexity of the SM, it should provide a computationally tractable method for learning high-level interaction features.

In the second iteration of the interaction matrix (abbreviated as SMconst and referred to as SM matrix[2] in Table 3), the SM coupling constants are used as fixed parameters: $g_Z = 0.758$ for the weak force among leptons, $g_s = 1.22$ for the strong force in jet interactions, and $g_e = 0.31$ for electromagnetic force in photon interactions. Interactions between jets and photons, as well as between b-jets and photons, are governed by the electromagnetic coupling constant $g_e$, as photons do not carry colour charge. Consequently, interactions are characterized as follows:

- For jet-$\gamma$ interactions, the coupling constant is modified to $g_e \times 0.5$ to reflect the assumption that jets originate mainly from quarks in the signals investigated in this work. The factor 0.5 is derived from the average quark charge, calculated as $\left(\frac{1}{3} + \frac{2}{3}\right)/2$, assuming an equal distribution of the quark charges of $\frac{1}{3}$ and $\frac{2}{3}$.

- For b-jet-$\gamma$ interactions, the electric charge of b-quarks is considered by using $g_e \times \frac{1}{3}$.

In the third interaction matrix (abbreviated as SM and referred to as SM matrix[3] in Table 3), the energy dependence of the coupling constants is taken into account:

- For Quantum Electrodynamics (QED), the running of the fine-structure constant $\alpha$ is described by the Renormalization Group Equation (RGE). At one-loop level for a given pair of particle types $(i, j)$, it can be approximated as:

$$\alpha(Q^2) = \frac{\alpha(\mu_0^2)}{1 - \frac{n\alpha(\mu_0^2)}{3\pi} \cdot \ln\left(\frac{Q^2}{\mu_0^2}\right)},$$

$$g_e = \sqrt{4\pi\alpha}. \tag{9}$$

The factor $n$ approximates the contribution of different particles in the loop. A value of $n = 3$ was used, considering only leptons. Other choices did not have much influence.

- For Quantum Chromodynamics (QCD), the running of $\alpha_s$ is more complex due to the non-Abelian nature of the theory. The one-loop RGE for a given pair of particle types $(i, j)$, $\alpha_s$ is:

$$\alpha_s(Q^2) = \frac{\alpha_s(\mu_0^2)}{1 + \frac{\alpha_s(\mu_0^2)(33-2n_f)}{12\pi} \ln\left(\frac{Q^2}{\mu_0^2}\right)},$$

$$g_s = \sqrt{4\pi\alpha_s}. \tag{10}$$

Here, $\mu_0 = 91.1876$ GeV, $\alpha(\mu_0) = \frac{1}{127.5}$, $\alpha_s(\mu_0) = 0.118$, and $n_f = 6$ is the number of active quark flavors at the energy scale $Q^2$. The constants were calculated at the

specific energy scale $Q^2 = \bar{p}_t^2 = \left(\frac{p_t^i + p_t^j}{2}\right)^2$, where $\bar{p}_t$ represents the average transverse momentum of a particle pair in an event. This energy scale was used in the RGEs to compute $\alpha(Q^2)$ and $\alpha_s(Q^2)$. $n_f = 6$ is kept constant throughout the calculations, even for energy scales below the top quark mass, to avoid additional complexity and maintain consistency across different energy scales. While varying $n_f$ based on the energy scale would provide a more precise treatment, the impact of this variation is expected to be minimal in the energy ranges relevant to this analysis.

The effective coupling strength for electromagnetic and strong interactions, $g_e$ and $g_s$, were calculated at a given energy scale $Q^2$, while $g_z$ was kept constant as in the previous version of the matrix.

The interaction matrix provides a structured approach to encoding SM-particle interactions for training machine learning models, especially models such as ParT. By simplifying the wide range of possible interactions into a prioritized scheme, the matrix allows learning to focus on the most important interactions. The interaction matrices are structured such that—for the transformer implementation—large negative numbers (e.g., -10k) are used if there is no interaction (i.e., masked). This masking is achieved by the softmax activation function, which exponentiates the values in the attention matrix, thereby pushing irrelevant values toward zero.

# 5 Results

## 5.1 Summary of model details

Table 3 summarizes the details of the machine learning (ML) models and their configurations as determined from the hyperparameter studies discussed above. In the following sections, the performance of these models will be discussed when applied to 4-top signal with different backgrounds and in the case of the top-top-Higgs signal.

## 5.2 A search for 4-top production

In order to investigate the relationship between the amount of training data and the model's performance, learning curves were plotted in Fig. 1, which shows the area under the ROC curve (AUC) scores as a function of training size. The x-axis represents the size of the training set, while the y-axis denotes the AUC score achieved by the model on a test set.

As illustrated in the figure, a clear trend of improving AUC scores with an increase in training set size is observed, affirming the hypothesis that larger datasets enhance model performance. Notably, this improvement is more pronounced in the initial stages of increasing data volume. However, beyond a certain point, the rate of improvement in AUC scores begins to plateau. This observation suggests that while additional training data is beneficial, the marginal gains in model accuracy diminish after reaching a certain dataset size.

Furthermore, the learning curves also provide insights into the data efficiency and learning capacity of the different models. Models such as PN and ParT demonstrate a steeper ascent in AUC scores with fewer data, indicating better data efficiency, while others show a more gradual improvement, reflecting their need for larger datasets to achieve comparable performance.

Both the graph PN and ParT models improve similarly with increasing training data, which may deviate from expectations due to either insufficient training data or a lack of feature richness, though the inclusion of physical interactions and pairwise features remains crucial for model performance.

Table 3: A summary of ML model details, including neural network (NN) structures and the respective loss functions used. Pairwise kinematic features are included in certain models. The particle input variables for these models are detailed in Table 2.

| NN structure | Pairwise kinematic features | Loss function |
|---|---|---|
| BDT | | |
| BDT$_{int.}$ | $m_{ij}, \Delta R_{ij}$ | |
| FCN | | |
| CNN | | |
| PN | | |
| PN$_{int.}$ | $m_{ij}, \Delta R_{ij}$ | |
| PN$_{int.\,SMids}$ | $m_{ij}, \Delta R_{ij} + $ SM matrix[1] | |
| PN$_{int.\,SM\,const}$ | $m_{ij}, \Delta R_{ij} + $ SM matrix[2] | Cross-entropy |
| PN$_{int.\,SM}$ | $m_{ij}, \Delta R_{ij} + $ SM matrix[3] | |
| ParT | | |
| ParT$_{int.}$ | $m_{ij}, \Delta R_{ij}$ | |
| ParT$_{int.\,SM\,(FL)}$ | $m_{ij}, \Delta R_{ij} + $ SM matrix[3] | Focal $[\alpha = 0.75, \gamma = 3]$ |
| ParT$_{int.\,SMids}$ | $m_{ij}, \Delta R_{ij} + $ SM matrix[1] | |
| ParT$_{int.\,SM\,const}$ | $m_{ij}, \Delta R_{ij} + $ SM matrix[2] | Cross-entropy |
| ParT$_{int.\,SM}$ | $m_{ij}, \Delta R_{ij} + $ SM matrix[3] | |
| SetT$_{int.\,SM}$ | $m_{ij}, \Delta R_{ij} + $ SM matrix[3] | |

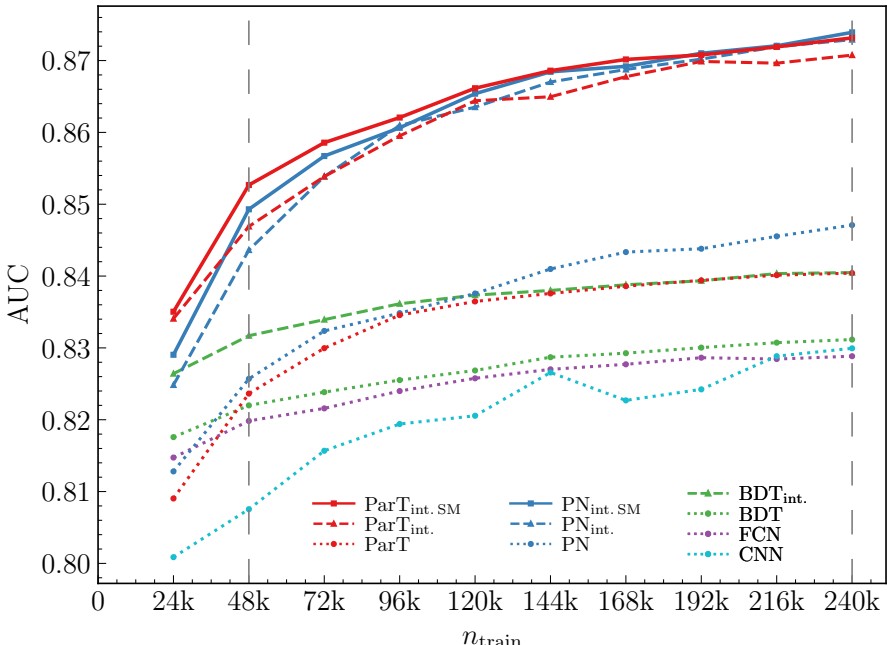

Figure 1: Learning curves of various machine learning models. This plot illustrates the relationship between the size of the training dataset and the AUC (Area Under the Curve) for each model. The vertical dashed lines represent the referenced training sizes of 48k and 240k event in the dataset.

In addition to the AUC learning curves, signal efficiencies at fixed background efficiencies of 30% and 70% were also analyzed. These results, which further demonstrate the models' performance, are discussed in Appendix A.1.

The analysis and key findings, based on the 48k training dataset, are summarized in Tables 4 and 5. Table 4 shows the AUC values along with the background efficiencies ($\epsilon_B$) at signal efficiencies ($\epsilon_S$)[3] of 30% and 70% for individual background processes and each evaluated method. Table 5 provides the overall results for all processes combined. In particular, the PN and ParT architectures, with the inclusion of pairwise features and SM running coupling constants (labelled 'int. SM'), consistently achieved the best performance across all metrics and at different signal efficiencies. The choice of $\epsilon_S = 0.7$ in Table 5 reflects a balance between high signal efficiency and manageable background levels, which is crucial for optimizing sensitivity in real-world experimental scenarios. At this signal efficiency, the models demonstrate their ability to preserve a significant fraction of signal events while effectively suppressing background contributions.

Comprehensive results covering other versions of the SM interaction matrices are presented in Table 8 in Appendix A.2.

The background efficiencies at a signal efficiency of 30% vary among the models. Generally, lower background efficiency at this signal efficiency level indicates a model's strength in maintaining signal detection while effectively rejecting a significant portion of the background. At a higher signal efficiency of 70%, background efficiencies increase for all models, which is expected as increasing signal efficiency typically comes at the cost of allowing more background events. Certain models, particularly $PN_{int.\,SM}$ and $ParT_{int.\,SM}$ with pairwise kinematic features and the SM interaction matrix[3], manage to maintain relatively lower background efficiencies, underscoring their efficiency in handling a more challenging balance between signal and background.

From both tables the PN and ParT architectures show improvements of up to 9% when comparing models labeled 'int.' (which include pairwise features) to models labeled 'int. SM' (which additionally incorporate the SM interaction matrix).

Full details covering the entire 240k training dataset are provided in Tables 7 and 9 in Appendix A.2.

Figure 2 presents an alternative metric that more effectively illustrates the significance of these differences. The y-axis shows the ratio of background rejection, defined as $1/\epsilon_B$, for various models relative to the PN baseline. This ratio quantifies how much better each model is at rejecting background events compared to the PN baseline. A value of 1 indicates that the model performs similarly to the baseline, whereas values above 1 indicate improved background rejection. For instance, values between 1.1 and 1.4 suggest a 10% to 40% improvement in background rejection compared to the baseline at signal efficiencies between 30% and 70%.

The performance of the 'int. SM' models relative to the PN baseline shows clear improvements for processes such as $t\bar{t} + h$, $t\bar{t} + WW$, and $t\bar{t} + Z$. However, the improvement is less pronounced for $t\bar{t} + W$, especially at lower signal efficiencies. This behavior can be attributed to the kinematic similarities between the 4-top signal and the $t\bar{t} + W$ background, particularly in terms of jet and lepton multiplicities. Both processes result in high jet multiplicities and can produce multiple isolated leptons in the final state, making it challenging for the models to distinguish between them. This is especially true at lower signal efficiencies, where models prioritize rejecting more distinguishable backgrounds. The complexity of the final states in processes such as $t\bar{t} + WW$, with additional jets and $W$ bosons, provides more features for the models to exploit, leading to better performance improvement compared to $t\bar{t} + W$.

Models incorporating pairwise kinematic features and the SM interaction matrix[3], such as $PN_{int.\,SM}$ and $ParT_{int.\,SM}$, consistently outperform the PN baseline. This improvement is particularly notable at signal efficiencies of 30% and 70% (indicated by the vertical dashed lines). These results suggest that including physical principles, such as those in the SM interaction matrix, significantly enhances the model's ability to reject background events.

---

[3]$\epsilon_S \equiv \frac{\text{TP}}{\text{TP+FN}}$, and $\epsilon_B \equiv \frac{\text{FP}}{\text{TN+FP}}$.

Table 4: The areas under the ROC curve and the background efficiencies at signal efficiencies of 70% and 30%, respectively, correspond to the 48k training dataset. The quoted uncertainties are extracted from three independent runs for each network architecture. Numbers in bold indicate the best performance. In cases where the performances of multiple architectures are the best within the uncertainty, the results are both indicated.

| | | BDT | BDT$_{\text{int.}}$ | FCN | CNN |
|---|---|---|---|---|---|
| | AUC | 0.825(0) | 0.831(0) | 0.821(2) | 0.778(6) |
| $t\bar{t}+h$ | $\epsilon_B(\epsilon_S=0.7)$ | 0.206(0) | 0.192(0) | 0.203(1) | 0.272(11) |
| | $\epsilon_B(\epsilon_S=0.3)$ | 0.026(1) | 0.026(0) | 0.026(1) | 0.037(1) |
| | AUC | 0.891(0) | 0.895(0) | 0.887(0) | 0.867(5) |
| $t\bar{t}+W$ | $\epsilon_B(\epsilon_S=0.7)$ | 0.099(0) | 0.092(0) | 0.103(1) | 0.125(8) |
| | $\epsilon_B(\epsilon_S=0.3)$ | 0.011(0) | 0.011(0) | 0.010(0) | 0.011(1) |
| | AUC | 0.740(0) | 0.746(0) | 0.737(1) | 0.745(2) |
| $t\bar{t}+WW$ | $\epsilon_B(\epsilon_S=0.7)$ | 0.347(0) | 0.339(0) | 0.342(5) | 0.335(3) |
| | $\epsilon_B(\epsilon_S=0.3)$ | 0.050(0) | 0.051(0) | 0.054(0) | 0.051(0) |
| | AUC | 0.833(0) | 0.856(0) | 0.836(0) | 0.839(1) |
| $t\bar{t}+Z$ | $\epsilon_B(\epsilon_S=0.7)$ | 0.191(0) | 0.163(0) | 0.192(0) | 0.190(4) |
| | $\epsilon_B(\epsilon_S=0.3)$ | 0.026(0) | 0.019(0) | 0.023(0) | 0.021(1) |
| | | **PN** | **PN$_{\text{int.}}$** | **PN$_{\text{int. SM}}$** | **ParT$_{\text{int. SM (FL)}}$** |
| | AUC | 0.824(0) | 0.842(1) | **0.846(1)** | 0.844(1) |
| $t\bar{t}+h$ | $\epsilon_B(\epsilon_S=0.7)$ | 0.199(0) | 0.176(3) | **0.171(2)** | 0.176(2) |
| | $\epsilon_B(\epsilon_S=0.3)$ | 0.025(0) | **0.019(1)** | 0.020(1) | 0.020(1) |
| | AUC | 0.887(0) | 0.895(2) | 0.900(1) | **0.902(4)** |
| $t\bar{t}+W$ | $\epsilon_B(\epsilon_S=0.7)$ | 0.102(1) | 0.097(1) | **0.091(1)** | **0.091(5)** |
| | $\epsilon_B(\epsilon_S=0.3)$ | 0.011(0) | 0.011(0) | **0.010(0)** | 0.011(0) |
| | AUC | 0.742(0) | 0.760(1) | 0.765(0) | 0.768(3) |
| $t\bar{t}+WW$ | $\epsilon_B(\epsilon_S=0.7)$ | 0.335(2) | 0.311(1) | 0.297(2) | 0.294(7) |
| | $\epsilon_B(\epsilon_S=0.3)$ | 0.051(0) | 0.044(1) | **0.044(1)** | 0.044(1) |
| | AUC | 0.851(0) | 0.879(1) | **0.887(1)** | 0.892(0) |
| $t\bar{t}+Z$ | $\epsilon_B(\epsilon_S=0.7)$ | 0.168(4) | 0.136(1) | **0.126(2)** | 0.119(4) |
| | $\epsilon_B(\epsilon_S=0.3)$ | 0.020(0) | 0.016(1) | 0.016(0) | 0.016(0) |
| | | **ParT** | **ParT$_{\text{int.}}$** | **ParT$_{\text{int. SM}}$** | **SetT$_{\text{int. SM}}$** |
| | AUC | 0.824(0) | 0.837(2) | **0.846(1)** | 0.845(1) |
| $t\bar{t}+h$ | $\epsilon_B(\epsilon_S=0.7)$ | 0.197(3) | 0.179(6) | 0.174(1) | 0.176(3) |
| | $\epsilon_B(\epsilon_S=0.3)$ | 0.023(0) | 0.020(0) | 0.020(0) | 0.020(0) |
| | AUC | 0.896(1) | 0.899(1) | **0.905(2)** | 0.898(1) |
| $t\bar{t}+W$ | $\epsilon_B(\epsilon_S=0.7)$ | 0.097(2) | 0.090(1) | **0.089(3)** | 0.094(2) |
| | $\epsilon_B(\epsilon_S=0.3)$ | 0.010(0) | 0.010(0) | **0.009(0)** | 0.011(0) |
| | AUC | 0.737(0) | 0.767(1) | **0.769(0)** | 0.763(1) |
| $t\bar{t}+WW$ | $\epsilon_B(\epsilon_S=0.7)$ | 0.354(3) | 0.295(5) | **0.288(2)** | 0.301(5) |
| | $\epsilon_B(\epsilon_S=0.3)$ | 0.050(1) | 0.040(0) | **0.042(0)** | 0.047(1) |
| | AUC | 0.839(1) | 0.885(0) | **0.891(1)** | 0.886(2) |
| $t\bar{t}+Z$ | $\epsilon_B(\epsilon_S=0.7)$ | 0.182(2) | 0.130(1) | **0.119(3)** | 0.129(4) |
| | $\epsilon_B(\epsilon_S=0.3)$ | 0.021(1) | 0.016(0) | **0.015(0)** | **0.014(0)** |

Table 5: Performance metrics for the 4-top signal, including the areas under the ROC curve and the background efficiencies at signal efficiencies of 70%. The results are based on the 48k training dataset. The quoted uncertainties are extracted from three independent runs for each network architecture. Numbers in bold indicate the best-performing models.

|  |  | **PN** | **PN$_{int.}$** | **PN$_{int. SM}$** |
|---|---|---|---|---|
| $t\bar{t}t\bar{t}$ | AUC | 0.8257(0) | 0.8436(1) | **0.8493(0)** |
|  | $\epsilon_B(\epsilon_S = 0.7)$ | 0.2015(0) | 0.1802(0) | **0.1717(0)** |
|  |  | **ParT** | **ParT$_{int.}$** | **ParT$_{int. SM}$** |
| $t\bar{t}t\bar{t}$ | AUC | 0.8236(0) | 0.8469(0) | **0.8527(0)** |
|  | $\epsilon_B(\epsilon_S = 0.7)$ | 0.2082(1) | 0.1739(2) | **0.1679(0)** |

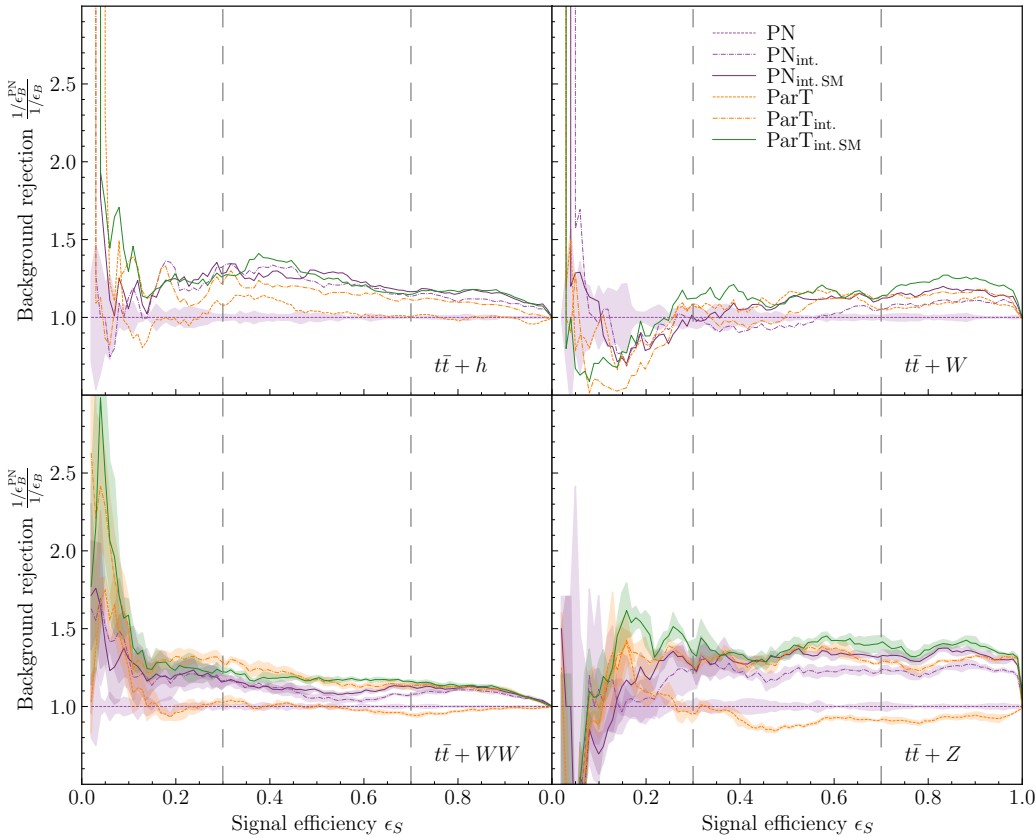

Figure 2: Signal efficiency versus background rejection plot for the four background processes corresponding to the 48k training dataset.

Fig. 3 displays the signal and background distributions as a function of the classifier score, normalized to the total cross-section. This figure, with solid lines and error bands, shows the mean and standard deviation observed over three independent runs for each architecture across the entire dataset. A critical observation here is the tendency of the best performing architectures to concentrate large portions of the background at lower classifier values, especially for background processes with higher cross-sections, such as $t\bar{t} + W$ and $t\bar{t} + Z$. This property is of crucial importance for the discrimination of backgrounds in signal fits in LHC experiments.

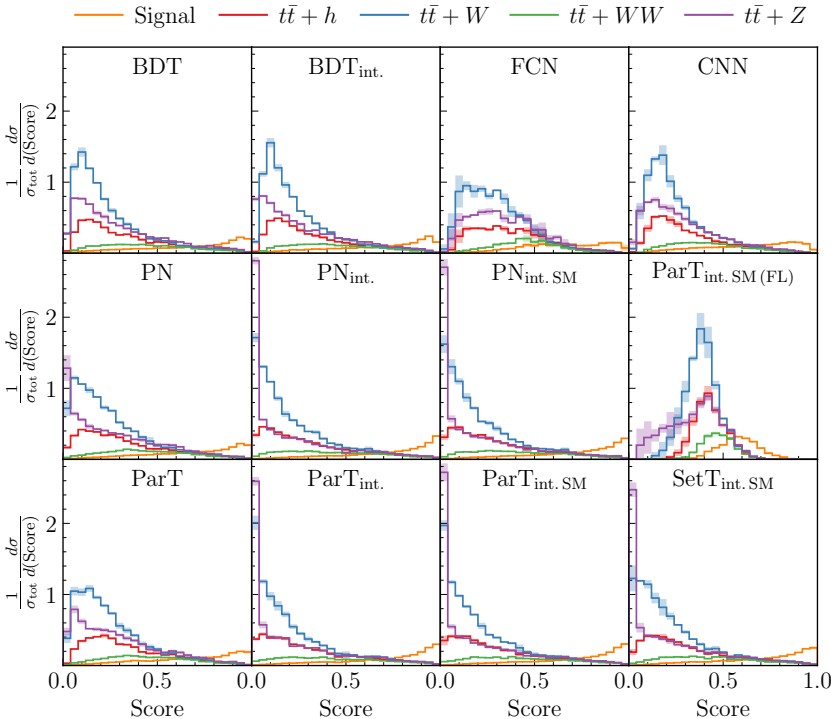

Figure 3: Distribution of classification scores for various models trained to distinguish signal and background processes. The histograms are normalized by the total cross-section, $\sigma_{\text{tot}}$, to represent the differential cross-section $\frac{1}{\sigma_{\text{tot}}}\frac{d\sigma}{d(\text{Score})}$. Here, $\sigma_{\text{tot}}$ is the sum of the cross-sections of the signal and all background processes. Signal events and four background processes ($t\bar{t}+h$, $t\bar{t}+W$, $t\bar{t}+WW$, and $t\bar{t}+Z$) are weighted by their respective cross-section. Shaded bands represent statistical uncertainties derived from the standard deviation of the scores across three independent runs of each model, reflecting the stability of each model's predictions on the full-size (240k) dataset. Each subplot corresponds to a specific model, highlighting differences in separation power and classification performance.

Table 6 compares the performance of various models at two distinct signal efficiency levels, $\epsilon_S = 0.3$ and $\epsilon_S = 0.7$. Significance ($\sigma$) is defined as the signal count ($s$) divided by the square root of the background count ($b$). This calculation is done under the assumption of a luminosity of $100\,\text{fb}^{-1}$ and incorporates the LO (Leading Order) cross-section from Table 1. It also includes the significance incorporating systematic uncertainties ($\sigma_{\delta_{\text{sys}}=0.2}$), where the effective background count ($b_{\text{sys}}$) is calculated as $b_{\text{sys}} = b + (b \cdot \delta_{\text{sys}})^2$ with $\delta_{\text{sys}} = 0.2$.

At $\epsilon_S = 0.3$, models capture 30% of true signal events. The significance without systematic uncertainties at this level suggests effective discrimination between signal and background. However, introducing a systematic uncertainties of 20% noticeably reduces the significance, underscoring the influence of factors such as instrumental or theoretical uncertainties. At $\epsilon_S = 0.7$, models identify 70% of true signal events. While this higher efficiency captures more signal events, the corresponding raw significance drops. The impact of systematic uncertainties is even more pronounced at this efficiency, as evidenced by the further decrease in $\sigma_{\delta_{\text{sys}}=0.2}$.

A comparative analysis reveals that the different versions of the particle transformer with the SM interaction matrix[3] (PartT$_{\text{int SM}}$ with and without focal loss, and as Set Transformer) achieve the highest significance without systematic errors at $\epsilon_S = 0.3$. In addition, ParT$_{\text{int. SM}}$ attains the highest significance when accounting for systematic errors at $\epsilon_S = 0.7$. These

Table 6: Significance values calculated for the entire 240k dataset.

|  |  | $\sigma$ | $\sigma_{\delta\mathbf{sys}=0.2}$ |
|---|---|---|---|
| BDT | $\epsilon_S = 0.3$ | 20.77 | 6.79 |
|  | $\epsilon_S = 0.7$ | 16.82 | 2.01 |
| BDT$_{\text{int.}}$ | $\epsilon_S = 0.3$ | 21.93 | 7.53 |
|  | $\epsilon_S = 0.7$ | 17.51 | 2.17 |
| FCN | $\epsilon_S = 0.3$ | 20.31 | 6.51 |
|  | $\epsilon_S = 0.7$ | 16.67 | 1.97 |
| CNN | $\epsilon_S = 0.3$ | 20.88 | 6.86 |
|  | $\epsilon_S = 0.7$ | 16.73 | 1.98 |
| PN | $\epsilon_S = 0.3$ | 23.09 | 8.29 |
|  | $\epsilon_S = 0.7$ | 17.68 | 2.21 |
| PN$_{\text{int.}}$ | $\epsilon_S = 0.3$ | 25.30 | 9.83 |
|  | $\epsilon_S = 0.7$ | **20.51** | **2.97** |
| PN$_{\text{int. SM}}$ | $\epsilon_S = 0.3$ | **25.65** | **10.09** |
|  | $\epsilon_S = 0.7$ | **20.50** | **2.97** |
| ParT | $\epsilon_S = 0.3$ | 22.37 | 7.82 |
|  | $\epsilon_S = 0.7$ | 17.72 | 2.23 |
| ParT$_{\text{int.}}$ | $\epsilon_S = 0.3$ | 24.54 | 9.29 |
|  | $\epsilon_S = 0.7$ | 20.21 | 2.89 |
| ParT$_{\text{int. SM}}$ | $\epsilon_S = 0.3$ | 25.36 | 9.88 |
|  | $\epsilon_S = 0.7$ | **20.53** | **2.98** |
| ParT$_{\text{int. SM (FL)}}$ | $\epsilon_S = 0.3$ | **26.19** | **10.48** |
|  | $\epsilon_S = 0.7$ | 20.28 | 2.91 |
| SetT$_{\text{int. SM}}$ | $\epsilon_S = 0.3$ | **25.58** | **10.03** |
|  | $\epsilon_S = 0.7$ | 20.18 | 2.88 |

findings highlight the crucial role of model selection based on specific analytical requirements and the significant impact of systematic errors, especially at higher signal efficiency levels.

Compared to the baseline graph network (PN), it is interesting to estimate how much the sample statistic (or integrated luminosity) would have to be increased in order to achieve a similar increase in significance, neglecting systematic errors. For instance, an increase in significance from 2.21 $\sigma$ (baseline PN model at 70% signal efficiency) to 2.98 $\sigma$ (ParT$_{\text{int. SM}}$) corresponds to an increase in integrated luminosity of approximately 82%. Similarly, an increase in significance from 8.29 $\sigma$ (baseline PN model at 30% signal efficiency) to 9.88 $\sigma$ (ParT$_{\text{int. SM}}$) corresponds to an increase in integrated luminosity of approximately 42%, and an increase from 8.29 $\sigma$ to 10.48 $\sigma$ (ParT$_{\text{int. SM (FL)}}$) corresponds to an increase in integrated luminosity of approximately 60%.

Finally, Fig. 4 shows the performance for the 48k training dataset using the AUC metric for PN as a function of $k$, the number of nearest neighbours. As one might expect, performance improves with $k$, eventually saturating when $k$ approaches the limit where every particle is connected to all other particles. It is important to note that this limit would require significantly more computational overhead in the context of jet physics, as the number of objects can grow

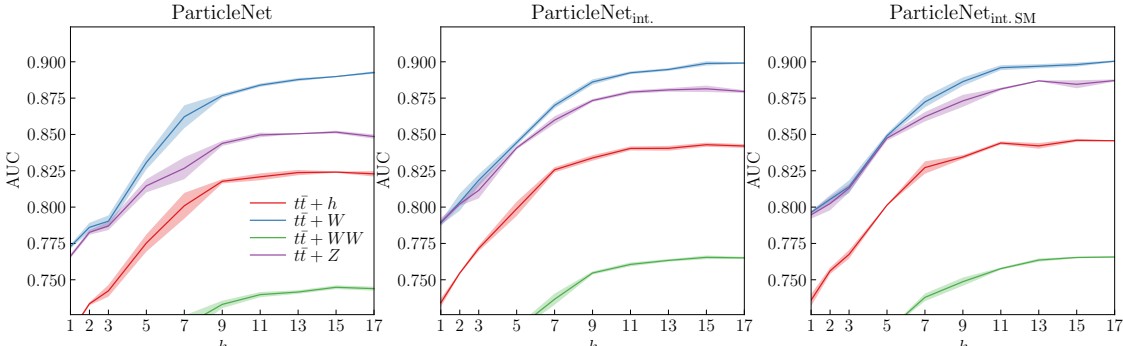

Figure 4: The performance of ParticleNet on the four background processes as a function of $k$, the number of nearest neighbours. The solid lines and error bands correspond to the mean and standard deviation over three independent runs for every value of $k$, corresponding to the 48k training dataset.

much larger and the complexity scales like $\mathcal{O}(kn)$. Here, the number of objects is limited, and setting $k = n$ is unproblematic. When $k$ approaches $n$, PN's graph becomes fully connected, enabling each particle to interact directly with all others. This setup effectively transforms PN into an architecture resembling the ParT, which uses self-attention mechanisms for global interactions. Consequently, both models exhibit similar performance due to their ability to model comprehensive particle relationships.

## 5.3 The top-top-Higgs searches

To evaluate the impact of including running coupling constants on the efficiency of neural networks, a second analysis was performed, focusing on the search for top-top-Higgs ($t\bar{t}+h$) signal. This analysis serves as a complementare study to the 4-top signal analysis, providing the confirmation about improvements of incorporating physical symmetries and pairwise features into ML models.

The dataset for the $t\bar{t}+h$ signal analysis is identical in composition to the dataset used for the 4-top signal analysis, except that the 4-top events are intentionally excluded. Detailed results covering all versions of the SM interaction matrices are presented in Table 10 in Appendix A.3.

These results confirm our earlier findings that integrating physical information into NN architectures enhance their ability to discriminate between signal and background events. The inclusion of the SM interaction matrix[3], in particular, leads to a significant reduction in background acceptance—by about 3% for PN and 3% for ParT compared to the models with only pairwise features—while maintaining high signal efficiency.

To complement these results, Table 11 in Appendix A.3 provides the significance estimates for the simplified $t\bar{t}+h$ analysis. It should be noted that not all backgrounds were included in this analysis (in comparison to a full ATLAS or CMS analysis), and the values should only be used to see how important background reduction can be. The reported values should therefore only be interpreted in terms of relative performance. For instance, an increase in significance from $3.80\,\sigma$ (baseline PN model at 70% signal efficiency) to $5.02\,\sigma$ ($\text{PN}_{\text{int. SM}}$) corresponds to an increase in integrated luminosity of about 75%, which confirms our previous findings.

# 6 Conclusions

In this work, a comprehensive comparison of event classifiers and a novel approach for event classification in particle physics was presented, where physical information, in particular energy-dependent SM interactions, is integrated into advanced machine learning models. This study focused on enhancing transformer models (ParT) with an attention matrix and graph networks, such as ParticleNet (PN) with edge features, both of which reflect the dynamical nature of SM interactions.

The results demonstrate that PN and ParT models achieve superior performance when pairwise features and SM interaction matrices are incorporated. Background suppression improved by 10–40% compared to baseline models without additional physical information, with up to 9% of this improvement directly attributable to the inclusion of SM interaction matrices, depending on the process and signal efficiency.

A simplified statistical analysis found that these machine learning models increase significance by up to 30% compared to the baseline model. Achieving a similar improvement in significance through increased luminosity ($L$) would require increasing $L$ by approximately 70%, assuming significance scales with $\sqrt{L}$ when both signal and background events are proportional to $L$.

It is concluded that embedding SM interactions as physical information in network structures is an important avenue in this field, which could lead to more accurate and efficient event classification in particle physics.

# Acknowledgments

**Funding information** The author(s) gratefully acknowledges the computer resources provided by Artemisa, funded by the European Union ERDF and the Comunitat Valenciana, as well as the technical support from the Instituto de Física Corpuscular, IFIC (CSIC-UV). R. RdA is supported by PID2020-113644GB-I00 from the Spanish Ministerio de Ciencia e Innovación and by PROMETEO/2022/69 from the Spanish GVA. RV is supported by the European Research Council (ERC) under the European Union's Horizon 2020 research and innovation programme (grant agreement No. 788223, PanScales).

# A Additional plots and tables

This appendix complements the main text by providing additional plots and comprehensive tables. The results for the entire 240k dataset are summarized, along with additional results for the 48k, providing a complete perspective on the scope of the data and the outcomes of the analysis.

## A.1 Signal efficiency at fixed Background efficiencies

The plots in Fig. 5 show the behaviour of signal efficiency ($\epsilon_S$) for different algorithms as a function of training data size. These are evaluated at fixed background efficiencies ($\epsilon_B$) of 30% and 70%. The goal of these plots is to observe how well each algorithm improves in signal efficiency as more training data is introduced, while maintaining a constant level of background rejection.

In Fig. 5a, which corresponds to a background efficiency of 30%, is it observed that both PN and ParT models show an increase in signal efficiency as the training size increases. The

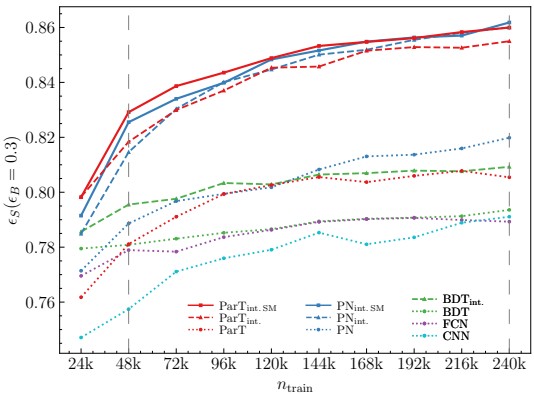

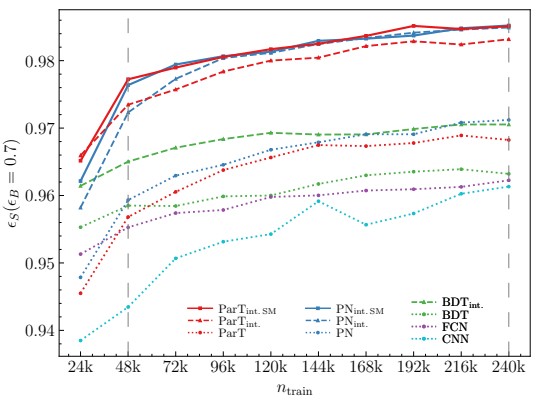

(a) Signal efficiency ($\epsilon_S$) at a fixed background efficiency ($\epsilon_B$) of 30%.

(b) Signal efficiency ($\epsilon_S$) at a fixed background efficiency ($\epsilon_B$) of 70%.

Figure 5: Signal efficiency as a function of training data size for each algorithm, evaluated at two fixed background efficiencies: (a) 30% and (b) 70%. These plots allow for comparison between different models' abilities to detect signal as the amount of training data increases.

inclusion of pairwise kinematic features and the SM interaction matrix in both models leads to a noticeable improvement. For instance, for larger training datasets, ParT with the SM interaction matrix consistently achieves a higher signal efficiency than models without it, highlighting the benefit of including physics-informed features in improving detection capabilities.

Similarly, in Fig. 5b, which shows results at a background efficiency of 70%, a similar pattern of improvement is observed, with the difference between models with and without the SM interaction matrix becoming more pronounced. This suggests that the introduction of such physical features particularly benefits scenarios with higher background levels. Both models exhibit improved signal efficiency with larger datasets, with ParT outperforming PN, especially at larger training sizes.

Comparing these figures to Fig. 1 in Section 5.2, which presents the AUC as a function of training size, a consistent trend is observed: as training data increases, both the AUC and signal efficiency improve for the same background rejection level. However, in the case of fixed background efficiency (Fig. 5), the focus shifts to how effectively the models can detect signal as more data is introduced, emphasizing the trade-off between signal efficiency and the level of background rejection.

## A.2 AUC

Figure 6 shows the ROC curves for all architectures against various backgrounds, providing a visual perspective on their comparative performances across the entire dataset for the $t\bar{t}t\bar{t}$ analysis.

Table 7 provides detailed results for the full 240k dataset from the $t\bar{t}t\bar{t}$ analysis, including the area under the ROC curve (AUC) and background efficiencies for various processes.

Tables 8 and 9 present performance metrics for the 4-top signal, including the areas under the ROC curve (AUC) and background efficiencies ($\epsilon_B$) at signal efficiencies of 70% ($\epsilon_S = 70\%$). Five scenarios are evaluated for both the PN and ParT models, demonstrating how the inclusion of pairwise features and Standard Model (SM) interaction matrices improves the performance of ML models in classifying signals. Table 8 provides detailed metrics for a 48k dataset, while table 9 summarizes results based on the full dataset (240k events). In both cases, the results consistently show that these additional features enhance model performance, with bold numbers highlighting the best-performing models.

Table 7: The areas under the ROC curve and the background efficiencies at signal efficiencies of 70% and 30%, respectively, correspond to the entire training dataset (240k events). The quoted uncertainties are extracted from three independent runs for each network architecture. Numbers in bold indicate the best performance. In cases where the performances of multiple architectures are the best within the uncertainty, the results are both indicated.

| | | **BDT** | **BDT**$_{\text{int.}}$ | **FCN** | **CNN** |
|---|---|---|---|---|---|
| $t\bar{t}+h$ | AUC | 0.833(0) | 0.840(0) | 0.832(0) | 0.838(3) |
| | $\epsilon_B(\epsilon_S=0.7)$ | 0.193(0) | 0.183(0) | 0.195(0) | 0.182(3) |
| | $\epsilon_B(\epsilon_S=0.3)$ | 0.022(0) | 0.022(0) | 0.023(1) | 0.021(2) |
| $t\bar{t}+W$ | AUC | 0.896(0) | 0.900(0) | 0.895(0) | 0.888(3) |
| | $\epsilon_B(\epsilon_S=0.7)$ | 0.093(0) | 0.087(0) | 0.093(1) | 0.107(3) |
| | $\epsilon_B(\epsilon_S=0.3)$ | 0.009(0) | 0.009(1) | 0.011(0) | 0.009(0) |
| $t\bar{t}+WW$ | AUC | 0.745(0) | 0.754(0) | 0.742(0) | 0.739(2) |
| | $\epsilon_B(\epsilon_S=0.7)$ | 0.339(0) | 0.317(2) | 0.341(0) | 0.344(2) |
| | $\epsilon_B(\epsilon_S=0.3)$ | 0.048(0) | 0.045(0) | 0.048(0) | 0.052(0) |
| $t\bar{t}+Z$ | AUC | 0.852(1) | 0.869(0) | 0.848(0) | 0.857(0) |
| | $\epsilon_B(\epsilon_S=0.7)$ | 0.167(1) | 0.149(2) | 0.170(2) | 0.161(2) |
| | $\epsilon_B(\epsilon_S=0.3)$ | 0.020(0) | 0.018(0) | 0.020(0) | 0.018(0) |
| | | **PN** | **PN**$_{\text{int.}}$ | **PN**$_{\text{int. SM}}$ | **ParT**$_{\text{int. SM (FL)}}$ |
| $t\bar{t}+h$ | AUC | 0.854(1) | **0.871(0)** | **0.872(0)** | 0.867(3) |
| | $\epsilon_B(\epsilon_S=0.7)$ | 0.161(2) | **0.129(1)** | **0.129(3)** | 0.138(4) |
| | $\epsilon_B(\epsilon_S=0.3)$ | 0.016(0) | 0.017(0) | 0.017(0) | **0.016(0)** |
| $t\bar{t}+W$ | AUC | 0.901(0) | 0.917(1) | **0.919(0)** | **0.918(1)** |
| | $\epsilon_B(\epsilon_S=0.7)$ | 0.089(1) | **0.072(1)** | **0.071(2)** | **0.071(2)** |
| | $\epsilon_B(\epsilon_S=0.3)$ | 0.008(0) | **0.007(0)** | **0.007(0)** | 0.007(0) |
| $t\bar{t}+WW$ | AUC | 0.759(2) | **0.791(0)** | **0.793(0)** | 0.791(3) |
| | $\epsilon_B(\epsilon_S=0.7)$ | 0.312(4) | 0.256(2) | 0.252(2) | **0.249(7)** |
| | $\epsilon_B(\epsilon_S=0.3)$ | 0.043(1) | 0.036(1) | 0.035(2) | 0.035(1) |
| $t\bar{t}+Z$ | AUC | 0.876(2) | **0.913(0)** | **0.913(0)** | 0.909(0) |
| | $\epsilon_B(\epsilon_S=0.7)$ | 0.139(5) | 0.095(1) | **0.094(2)** | 0.097(1) |
| | $\epsilon_B(\epsilon_S=0.3)$ | 0.015(0) | 0.013(0) | 0.012(0) | **0.010(0)** |
| | | **ParT** | **ParT**$_{\text{int.}}$ | **ParT**$_{\text{int. SM}}$ | **SetT**$_{\text{int. SM}}$ |
| $t\bar{t}+h$ | AUC | 0.843(1) | 0.869(0) | **0.871(0)** | 0.864(3) |
| | $\epsilon_B(\epsilon_S=0.7)$ | 0.179(3) | 0.131(2) | 0.132(0) | 0.141(4) |
| | $\epsilon_B(\epsilon_S=0.3)$ | 0.019(0) | **0.015(0)** | **0.015(0)** | **0.016(1)** |
| $t\bar{t}+W$ | AUC | 0.901(0) | 0.915(0) | **0.918(1)** | 0.915(2) |
| | $\epsilon_B(\epsilon_S=0.7)$ | 0.087(3) | 0.078(1) | **0.072(1)** | 0.074(2) |
| | $\epsilon_B(\epsilon_S=0.3)$ | 0.008(0) | 0.009(0) | 0.008(0) | 0.009(0) |
| $t\bar{t}+WW$ | AUC | 0.753(1) | **0.792(1)** | **0.792(1)** | 0.786(2) |
| | $\epsilon_B(\epsilon_S=0.7)$ | 0.318(5) | 0.250(2) | **0.248(2)** | 0.257(5) |
| | $\epsilon_B(\epsilon_S=0.3)$ | 0.047(1) | **0.032(0)** | 0.034(0) | 0.036(1) |
| $t\bar{t}+Z$ | AUC | 0.866(0) | 0.907(1) | **0.912(0)** | 0.907(2) |
| | $\epsilon_B(\epsilon_S=0.7)$ | 0.150(2) | 0.098(2) | **0.093(3)** | 0.100(4) |
| | $\epsilon_B(\epsilon_S=0.3)$ | 0.017(0) | 0.012(1) | **0.011(0)** | **0.011(0)** |

## A.3 tth

Detailed results for all versions of the SM interaction matrices are presented in Table 10, which provides performance metrics for the top-top-Higgs ($t\bar{t}+h$) signal. The table reports the areas under the ROC curve (AUC) as well as the background efficiencies ($\epsilon_B$) at signal efficiency of $\epsilon_S = 70\%$. It is important to note that although the full dataset was employed for the 4-top signal analysis, the same dataset, excluding the 4-top data, was used for the top-top-Higgs signal analysis.

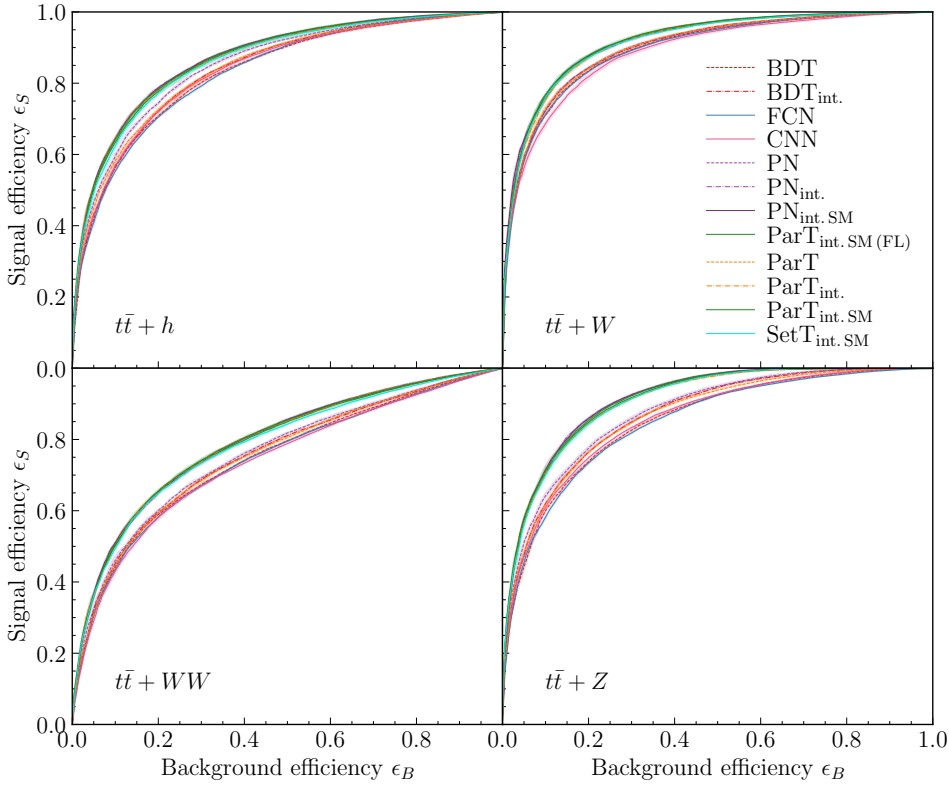

Figure 6: Receiver Operating Characteristic (ROC) curves for all architectures for the 4-top signal across the four background processes. The solid lines and error bands represent the mean and standard deviation from three independent runs for each architecture over the entire 240k training dataset.

Table 8: Performance metrics for the 4-top signal, including the areas under the ROC curve and the background efficiencies at signal efficiencies of 70%. The results are based on the 48k training dataset. The quoted uncertainties are extracted from three independent runs for each network architecture. Numbers in bold indicate the best performance.

| | | PN | PN$_{\text{int.}}$ | PN$_{\text{int. SMids}}$ | PN$_{\text{int. SM const}}$ | PN$_{\text{int. SM}}$ |
|---|---|---|---|---|---|---|
| $t\bar{t}t\bar{t}$ | AUC | 0.8257(0) | 0.8436(1) | 0.8468(0) | 0.8474(0) | **0.8493(0)** |
| | $\epsilon_B(\epsilon_S = 0.7)$ | 0.2015(0) | 0.1802(0) | 0.1756(3) | 0.1753(2) | **0.1717(0)** |
| | | ParT | ParT$_{\text{int.}}$ | ParT$_{\text{int. SMids}}$ | ParT$_{\text{int. SM const}}$ | ParT$_{\text{int. SM}}$ |
| $t\bar{t}t\bar{t}$ | AUC | 0.8236(0) | 0.8469(0) | 0.8441(1) | 0.8464(2) | **0.8527(0)** |
| | $\epsilon_B(\epsilon_S = 0.7)$ | 0.2082(1) | 0.1739(2) | 0.1793(1) | 0.1743(3) | **0.1679(0)** |

Table 9: Performance metrics for the 4-top signal, including the areas under the ROC curve and the background efficiencies at signal efficiencies of 70%. The results are based on the entire 240k training dataset. The quoted uncertainties are extracted from three independent runs for each network architecture. Numbers in bold indicate the best-performing models.

|  |  | PN | PN$_{\text{int.}}$ | PN$_{\text{int. SMids}}$ | PN$_{\text{int. SM const}}$ | PN$_{\text{int. SM}}$ |
|---|---|---|---|---|---|---|
| $t\bar{t}t\bar{t}$ | AUC | 0.8471(1) | 0.8729(0) | 0.8725(0) | 0.8727(0) | **0.8739(0)** |
|  | $\epsilon_B(\epsilon_S = 0.7)$ | 0.1758(3) | 0.1387(1) | 0.1377(0) | 0.1384(0) | **0.1369(1)** |
|  |  | ParT | ParT$_{\text{int.}}$ | ParT$_{\text{int. SMids}}$ | ParT$_{\text{int. SM const}}$ | ParT$_{\text{int. SM}}$ |
| $t\bar{t}t\bar{t}$ | AUC | 0.8404(0) | 0.8708(0) | 0.8715(0) | 0.8717(0) | **0.8732(0)** |
|  | $\epsilon_B(\epsilon_S = 0.7)$ | 0.1842(3) | 0.1394(0) | 0.1389(2) | 0.1372(1) | **0.1366(0)** |

Table 10: Performance metrics for the top-top-Higgs signal, including the areas under the ROC curve and the background efficiencies at signal efficiencies of 70%. The results are based on the entire 240k training dataset. The quoted uncertainties are extracted from three independent runs for each network architecture. Numbers in bold indicate the best-performing models.

|  |  | PN | PN$_{\text{int.}}$ | PN$_{\text{int. SMids}}$ | PN$_{\text{int. SM const}}$ | PN$_{\text{int. SM}}$ |
|---|---|---|---|---|---|---|
| $t\bar{t} + h$ | AUC | 0.8146(2) | 0.8505(0) | 0.8489(1) | 0.8505(0) | **0.8523(0)** |
|  | $\epsilon_B(\epsilon_S = 0.7)$ | 0.2292(1) | 0.1787(0) | 0.1785(1) | 0.1764(3) | **0.1733(1)** |
|  |  | ParT | ParT$_{\text{int.}}$ | ParT$_{\text{int. SMids}}$ | ParT$_{\text{int. SM const}}$ | ParT$_{\text{int. SM}}$ |
| $t\bar{t} + h$ | AUC | 0.8058(1) | 0.8507(0) | 0.8473(0) | 0.8497(0) | **0.8532(0)** |
|  | $\epsilon_B(\epsilon_S = 0.7)$ | 0.2399(2) | 0.1794(1) | 0.1836(3) | 0.1801(1) | **0.1748(1)** |

Five scenarios are provided for both the PN and the ParT models: the standard ParT/PN architecture, ParT/PN with the inclusion of pairwise features (int.), ParT/PN with the first iteration of the interaction matrix (SMids), ParT/PN with the second iteration of the interaction matrix (where the SM coupling constants are fixes parameters) and ParT/PN with the inclusion of SM running coupling constants (int. SM, representing the third iteration).

These results show how adding pairwise features and SM interaction matrices improves ML models for classifying signals. The gradual improvement of the interaction matrix—from basic pairwise features to fixed and running coupling constants—leads to consistent progress in both AUC and background rejection. Using the most advanced setup (int. SM) allows the models to perform the best, showing the importance of including physical principles in ML models for HEP.

Table 11 provides the significance estimates for the simplified $t\bar{t} + h$ analysis.

Table 11: Significance values calculated for the top-top-Higgs signal.

|  |  | $\sigma$ | $\sigma_{\delta\text{sys}=0.2}$ |
|---|---|---|---|
| PN | $\epsilon_S = 0.3$ | 32.18 | 7.63 |
|  | $\epsilon_S = 0.7$ | 34.30 | 3.80 |
| PN$_{\text{int.}}$ | $\epsilon_S = 0.3$ | **37.53** | **10.27** |
|  | $\epsilon_S = 0.7$ | 38.75 | 4.84 |
| PN$_{\text{int. SM}}$ | $\epsilon_S = 0.3$ | **37.86** | **10.44** |
|  | $\epsilon_S = 0.7$ | **39.50** | **5.02** |
| ParT | $\epsilon_S = 0.3$ | 30.24 | 6.76 |
|  | $\epsilon_S = 0.7$ | 33.48 | 3.62 |
| ParT$_{\text{int.}}$ | $\epsilon_S = 0.3$ | 36.82 | 9.90 |
|  | $\epsilon_S = 0.7$ | 38.39 | 4.75 |
| ParT$_{\text{int. SM}}$ | $\epsilon_S = 0.3$ | **37.20** | **10.10** |
|  | $\epsilon_S = 0.7$ | **39.05** | **4.91** |

# B   Networks optimization

Hyperparameter optimization is a crucial step in machine learning research. Conventional methods, such as GridSearch [46], are often inefficient and time-consuming. Therefore, Optuna [47] was chosen for this process.

Optuna implements a *define-by-run* approach, where an objective function is defined and optimized. This function is built using trial objects, and during each trial, the function samples values for hyperparameters from predefined ranges. Unlike GridSearch, which requires strictly defined search spaces, Optuna allows the search spaces to be constructed dynamically during the optimization process. Sampling is performed using Tree-structured Parzen Estimators (TPE) [48]. The model is then trained using the sampled parameters, and the trial is evaluated based on a defined score—typically a validation metric. In our case, this score was maximized to improve model performance.

The models were trained on different hardware setups, and although they were not benchmarked, they can be categorized by the time required for training. FCN and CNN models were the least time-consuming, whereas ParticleNet and Particle Transformer models required the most time, largely due to their inclusion of pairwise interactions and SM couplings. The training speed is highly dependent on the hardware used.

Below is a summary of the hyperparameters, the ranges used during optimization, and the optimal values found. When a range is preceded by `logarithmic`, it indicates that sampling was performed over a logarithmic domain.

## B.1 LightGBM

The optimized hyperparameters for the LightGBM model are shown in Table 12.

Table 12: The optimized hyperparameters for the LightGBM model included `n_estimators` (the number of boosted trees to fit), `num_leaves` (the maximum number of leaves per tree), `max_depth` (the maximum depth of each tree), `subsample` (the fraction of data used per tree), and the learning rate.

| Hyperparameter | Optimization Range | Optimal Value |
|---|---|---|
| learning_rate | $[1e^{-1}, 1e^{-2}, 1e^{-3}, 1e^{-4}]$ | $1e^{-3}$ |
| num_leaves | $[100, 200, \cdots, 1000]]$ | 500 |
| max_depth | $[5, 6, \cdots, 20]$ | 15 |
| subsample | $[0.5, 0.6, \cdots, 0.9]$ | 0.8 |
| n_estimators | $[100, 200, \cdots, 10000]$ | 5000 |

## B.2 FCN

The optimized hyperparameters for the FCN model are provided in Table 13.

Table 13: The hyperparameters optimized for the FCN. The table shows the range of values explored during the hyperparameter search and the optimal values found for each parameter. The hyperparameters include the learning rate, batch size, number of hidden layers (`n_layers`), and the number of neurons in each layer (`n_units_l`$i$), where $i$ represents the index of the dense layer. The number of neurons in each layer was optimized on a logarithmic scale.

| Hyperparameter | Optimization Range | Optimal Value |
|---|---|---|
| learning_rate | $[1e^{-1}, 1e^{-2}, 1e^{-3}, 1e^{-4}]$ | $1e^{-3}$ |
| batch_size | $[32, 64, 128, 256, 512]$ | 32 |
| n_layers | $[1, 2, \cdots, 15]$ | 13 |
| n_units_l0 | | 514 |
| n_units_l1 | | 432 |
| n_units_l2 | | 289 |
| n_units_l3 | | 320 |
| n_units_l4 | | 1017 |
| n_units_l5 | | 53 |
| n_units_l6 | logarithmic$[16, 1024]$ | 16 |
| n_units_l7 | | 249 |
| n_units_l8 | | 76 |
| n_units_l9 | | 91 |
| n_units_l10 | | 85 |
| n_units_l11 | | 145 |
| n_units_l12 | | 37 |

## B.3 CNN

The optimized hyperparameters for the CNN model are detailed in Table 14.

Table 14: The hyperparameters optimized for the 1D CNN. The table shows the range of values explored during the hyperparameter search and the optimal values found for each parameter. These include the number of convolutional layers in the first block (`n_conv_layers1`) and the second block (`n_conv_layers2`), the number of filters (`filter_ln`) and kernel sizes (`kernel_ln`) for each convolutional layer, where $n$ is the index of the layer in the convolutional block. It also includes the number of fully connected (dense) layers (`n_layers`) and the number of neurons in each dense layer (`n_units_li`), where $i$ represents the index of the dense layer. The batch size (`batch_size`) was optimized on a logarithmic scale.

| Hyperparameter | Optimization Range | Optimal Value |
|---|:---:|:---:|
| learning_rate | $[1e^{-1}, 1e^{-2}, 1e^{-3}, 1e^{-4}]$ | $1e^{-3}$ |
| batch_size | $[32, 64, 128, 256, 512]$ | 256 |
| n_conv_layers1 | $[1, 2, \cdots, 5]$ | 4 |
| n_conv_layers2 | | 2 |
| filter_l0 | | 16 |
| filter_l1 | | 8 |
| filter_l2 | $[8, 16, 32, 64]$ | 16 |
| filter_l3 | | 32 |
| filter_l4 | | 8 |
| filter_l5 | | 32 |
| kernel_l0 | | 2 |
| kernel_l1 | | 2 |
| kernel_l2 | $[1, 2, \cdots, 10]$ | 2 |
| kernel_l3 | | 5 |
| kernel_l4 | | 3 |
| kernel_l5 | | 3 |
| n_layers | $[1, 2, \cdots, 15]$ | 2 |
| n_units_l0 | logarithmic$[16, 1024]$ | 128 |
| n_units_l1 | | 16 |

## B.4 PN

Table 15 presents the optimized hyperparameters used for the ParticleNet (PN) model. Each node connects to its $k = 13$ nearest neighbors for local feature aggregation. Key hyperparameters include a learning rate of 0.001 and batch size of 512. The architecture uses three Edge-Conv blocks with channels $(64, 64, 64), (128, 128, 128), (256, 256, 256)$ to capture progressively complex particle interactions. Following these, the fully connected layers—configured as $(64, 0.1), (256, 0.1), (64, 0.1)$ with dropout rates of 0.1—process and refine the features, where each layer containing 64, 256, and 64 units, respectively. The auxiliary fully connected layers consist of two layers, each with 32 units and a dropout rate of 0.1, to provide additional feature processing. Pair embedding and attention dimensions of $[64, 64, 64]$ enable the model to capture pairwise relationships and dynamically weigh the importance of neighboring particles, enhancing feature aggregation. Auxiliary dimensions $E_T^{\text{miss}}$ and $\phi_{E_T^{\text{miss}}}$ in the final layer further contribute to classification performance.

Table 15: The hyperparameters optimized for the PN include the learning rate, batch size, convolutional layer dimensions (`conv_dim`), fully connected layer configurations (`fc_dim`), and the number of nearest neighbors ($k$) considered for each particle in the dynamic graph construction. Additional layers for embedding pairwise features (`pair_embeddims`) and auxiliary fully connected layers (`aux_fc_params`) provide enhanced feature extraction and processing, while attention dimensions (`attention_dims`) enable selective focusing on relevant particle interactions, improving background rejection.

| Hyperparameter | Optimal Value |
|---|---:|
| `learning_rate` | $1e^{-3}$ |
| `batch_size` | 512 |
| `k` | 13 |
| `conv_dim` | [(64, 64, 64), (128, 128, 128), (256, 256, 256)] |
| `fc_dim` | [(64, 0.1), (256, 0.1), (64, 0.1)] |
| `pair_embed_dims` | |
| `attention_dims` | [64, 64, 64] |
| `aux_fc_params` | [(32, 0.1), (32, 0.1)] |

## B.5 ParT

The optimized hyperparameters for the ParT model are listed in Table 16. The model's learning rate is set to 0.001, with a batch size of 512, ensuring a balance between computational efficiency and stable training. The architecture employs 8 Transformer layers (`num_layers`), allowing the model to capture hierarchical dependencies in particle interactions through self-attention mechanisms. Embedding dimensions (`embed_dims`) are set to [128, 512, 128], providing flexibility in representation, while pairwise embedding dimensions (`pair_embed_dims`), set to [64, 64, 64] if pairwise features are enabled, allow the model to capture interactions between pairs of particles.

The fully connected network (FCN) consists of three layers with node configurations of [64, 256, 64], each followed by a dropout rate of 0.1, transforming features extracted by the Transformer layers and contributing to the final classification. The auxiliary fully connected layers (`aux_fc_params`), with configurations [(32, 0.1), (32, 0.1)], where 0.1 is the dropout rate, further refine the model's representation by integrating event-level features, specifically $E_T^{\text{miss}}$ (missing transverse energy) and $\phi_{E_T^{\text{miss}}}$.

Table 16: The hyperparameters optimized for the ParT model. The model's configuration includes various embedding dimensions, layer counts, and fully connected layer specifications. The pairwise embedding dimensions (`pair_embed_dims`) are included only if pairwise features are incorporated, enabling the model to capture interactions between particles.

| Hyperparameter | Optimal Value |
|---|---:|
| `learning_rate` | $1e^{-3}$ |
| `batch_size` | 512 |
| `num_layers` | 8 |
| `embed_dims` | [128, 512, 128] |
| `pair_embed_dims` | [64, 64, 64] (if enabled) |
| `fc_params` | [(64, 0.1), (256, 0.1), (64, 0.1)] |
| `aux_fc_params` | [(32, 0.1), (32, 0.1)] |

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
