# Peer review of "Attention to the strengths of physical interactions: Transformer and graph-based event classification for particle physics experiments"

_SciPost Physics, doi:SciPost Phys. 19, 028 (2025)_

## Round 3 · Referee Report · Anonymous (Referee 1) · 2025-3-3

Report

The authors addressed all the previous comments and I support the publication of the updated paper.

Recommendation

Publish (meets expectations and criteria for this Journal)

---

## Round 3 · Referee Report · Tilman Plehn (Referee 3) · 2025-3-11

Report

Thank you for considering my questions and comments.

Recommendation

Publish (easily meets expectations and criteria for this Journal; among top 50%)

---

## Round 3 · Referee Report · Anonymous (Referee 2) · 2025-3-31

Report

I thank the authors for carefully addressing my comments and answering my questions. However, I need to follow up on point 5 from my original review:

Original point 5) Conclusions: “10% of the improvement directly attributable to the SM interaction matrix.” - If I compare all background rejections of “int.” and “int. SM” in Table 6, I do not see in any of the numbers an improvement near 10%. The same is true for the background rejections in Table 4. Or do you mean that it is 10% of the overall 10-40% improvement, i.e. only 1-4%? If this is the case, please rephrase the conclusions and the abstract for more clarity.

Authors’ answer: The number 10 percent comes from the results of table 4 and 5. Our baseline was the signal efficincy of 70 percent. Overall (Table 5) the background is reduced e.g. for the 4-top case from 0.18 to 0.17 for overall (i.e. 6 percent absolute) and e.g. from 0.13 to 0.119 for the Z+jets sample (i.e. 9 percent absolute (Table 4). We changed the conclusion to “...approximately up to 10% of this improvement..”.

Follow-up: I am afraid I am not fully convinced by the answer. I acknowledge the more careful phrasing in Section 5.2 (“improvements of up to approximately 10%”), but the statements in the abstract and in the conclusions are still stronger (“with an additional gain of approximately 10% (absolute) due to the SM interaction matrix” and “with approximately 10% (absolute) of this improvement directly attributable to the inclusion of SM interaction matrices”). Moreover, I am afraid that I still find the statement about the improvements exaggerated and I apologize if I miss a point here. Following the authors’ argument to focus on improvements in background rejection at a signal efficiency of 70%, I calculate the following improvements from Tables 4 and 5:

Table 4 (comparing PN_int to PN_int.SM and ParT_int to ParT_int.SM for 70% efficiency): PN (ttH): 2.8% PN (ttW): 6.2% PN (ttWW): 4.5% PN (ttZ): 7.4% ParT (ttH): 2.8% ParT (ttW): 1.1% ParT (ttWW): 2.4% ParT (ttZ): 8.5%

Table 5: PN (4top): 4.7% ParT (4top): 3.5%

None of these improvements is as large as 10% and in total, there is an average improvement of 4.4%. I personally would think that a statement of “approximately 5% improvement" would better reflect the numbers in these tables. I ask the authors to consider this.

In addition, I would like to point out two minor aspects that I trust the authors to take care of:

  • Introduction, paragraph 2: Please double-check Refs. [6-10] and [11-15]. It seems that they have been mixed up, i.e. [6-10] seems to also include event-level references and [11-15] are only the jet references, while the text says otherwise.

  • Conclusions, last sentence: “It is concluded that embedding SM interactions as physical information in network structures represents (?) an important avenue…” (typo?)

Recommendation

Ask for minor revision

  • validity: -
  • significance: -
  • originality: -
  • clarity: -
  • formatting: -
  • grammar: -

Author:  Polina Moskvitina  on 2025-04-04  [id 5338]

(in reply to Report 3 on 2025-03-31)
Category:
answer to question

Dear Referee,

We would like to thank you for your feedback on our updated manuscript.
Please find attached a PDF file containing our point-by-point responses to your comments.

Attachment:

ReplyToReferee_Round2.pdf

---

## Round 3 · Author Response

We thank the referees for their feedback. We have carefully addressed all major and minor comments to improve the clarity, quality, and presentation of our manuscript. Specifically, we have added detailed explanations of key methods, expanded discussions on the inclusion of Standard Model (SM) interaction matrices, and revised figures and tables for better accessibility.

---

## Round 3 · List of Changes

1. Abstract and Introduction
  2. Revised the abstract and introduction to clearly highlight the novelty of the proposed method, particularly the use of Standard Model (SM) interaction matrices for event classification.
  3. Clarified the broader scientific context and relevance of the proposed methodology for event-level classification and high-energy physics (HEP) data analysis.
  4. Added citations to relevant publications on event-level classifiers and neural networks in HEP, including works from ATLAS, CMS, ALICE, and LHCb collaborations.

  5. Methods

  6. Expanded descriptions of attention mechanisms in Transformer models (Sections 3.5 and 3.6) to differentiate between graph networks and Transformer-based architectures.
  7. Improved the explanation of the Binary Cross-Entropy (BCE) loss and focal loss in Section 3.7, including definitions for all symbols to facilitate implementation by readers.
  8. Clarified the role of pairwise kinematic features and the Standard Model Interaction Matrix (Section 4.1) and distinguished between covariant representations and physics-specific information.

  9. Figures and Tables

  10. Figure 1: Updated curve labels for better clarity, using distinct line styles and colors as suggested by the referees.
  11. Figure 2: Added explanations for the observed differences in performance for various models.
  12. Table 4 and Table 5: Revised descriptions to better explain background rejection improvements and the contribution of SM interaction matrices.
  13. Corrected typos in figure captions and legends, such as "FNC" → "FCN."
  14. Clarified the meaning of variables in tables, including the "γtag" variable in Table 2.

  15. Results and Discussions

  16. Addressed the discrepancy between Particle Transformer performance with and without physics-specific features (Fig. 1).
  17. Clarified why BDTs performed well despite their simplicity, compared to ParticleNet and Transformer models with no pairwise features.
  18. Added a discussion on how performance scales with additional training data and how this is influenced by dataset richness and physics-informed features (Section 5.2).
  19. Revised Appendix B to include detailed explanations of hyperparameter optimization processes for all models.

  20. Conclusions

  21. Rephrased the conclusion to emphasize the specific contribution of SM interaction matrices, quantifying their role in background rejection and signal significance improvements.

  22. References

  23. Included additional references to foundational works on convolutional neural networks (CNNs) and dropout mechanisms.
  24. Cited the "HEPML Living Review" for relevant literature on ML applications in HEP.

  25. Minor Revisions

  26. Clarified the statement on dataset size and balancing to address apparent inconsistencies with Table 1.
  27. Corrected typos, such as "ongoing coupling constants" → "running coupling constants."
  28. Clarified terminology and added missing explanations for technical details.

---

## Editorial Decision

published